# Regulation of ER Composition and Extent, and Putative Action in Protein Networks by ER/NE Protein TMEM147

**DOI:** 10.3390/ijms221910231

**Published:** 2021-09-23

**Authors:** Giannis Maimaris, Andri Christodoulou, Niovi Santama, Carsten Werner Lederer

**Affiliations:** 1Department of Biological Sciences, University of Cyprus, Nicosia 1678, Cyprus; maimaris.giannis@ucy.ac.cy (G.M.); christodoulou.c.andri@ucy.ac.cy (A.C.); santama@ucy.ac.cy (N.S.); 2Department of Molecular Genetics Thalassaemia, The Cyprus Institute of Neurology and Genetics, Nicosia 2371, Cyprus

**Keywords:** nuclear envelope, endoplasmic reticulum, network analysis, gene ontology, RNA interference

## Abstract

Nuclear envelope (NE) and endoplasmic reticulum (ER) collaborate to control a multitude of nuclear and cytoplasmic actions. In this context, the transmembrane protein TMEM147 localizes to both NE and ER, and through direct and indirect interactions regulates processes as varied as production and transport of multipass membrane proteins, neuronal signaling, nuclear-shape, lamina and chromatin dynamics and cholesterol synthesis. Aiming to delineate the emerging multifunctionality of TMEM147 more comprehensively, we set as objectives, first, to assess potentially more fundamental effects of TMEM147 on the ER and, second, to identify significantly TMEM147-associated cell-wide protein networks and pathways. Quantifying curved and flat ER markers RTN4 and CLIMP63/CKAP4, respectively, we found that TMEM147 silencing causes area and intensity increases for both RTN4 and CLIMP63, and the ER in general, with a profound shift toward flat areas, concurrent with reduction in DNA condensation. Protein network and pathway analyses based on comprehensive compilation of TMEM147 interactors, targets and co-factors then served to manifest novel and established roles for TMEM147. Thus, algorithmically simplified significant pathways reflect TMEM147 function in ribosome binding, oxidoreductase activity, G protein-coupled receptor activity and transmembrane transport, while analysis of protein factors and networks identifies hub proteins and corresponding pathways as potential targets of TMEM147 action and of future functional studies.

## 1. Introduction

Transmembrane Protein 147 (TMEM147), also known as NIFIE14, is a small, fairly recently discovered protein that localizes to the endoplasmic reticulum (ER) [1,2]. Encoded on the large arm of chromosome 19 (19q13.12) in *Homo sapiens*, it consists of seven exons, which through alternative splicing give rise to multiple isoforms [3], three of which have been validated as variants for the NCBI Gene database (Gene ID: 10430) [4]. The most abundant of the latter three in-frame variants is also the largest and translates to a 224-amino-acid (aa) protein of 26.2 kDa (see Table 1 and Appendix A).

TMEM147 has seven transmembrane domains, each of which fully span the ER membrane, with the N-terminus facing the ER lumen and the C-terminus facing the cytosol [5]. Its expression in many tissues [6] (see Appendix A) suggests its general and wide-ranging physiological role, as does the high level of conservation of the *TMEM147* gene across mammals, with sequence similarity in human, mouse, rat and bovine species of up to 99% [1,2]. However, *TMEM147* is phylogenetically older than the class of mammals and was originally isolated in zebrafish. There, early embryonic development is controlled by the NODAL signal transduction pathway, which Haffner and co-workers found to be modulated by a protein complex containing Nicalin and NOMO [7]. Differences in molecular weight between the native protein complex (200–220 kDa) and its known denatured components, Nicalin (60 kDa) and NOMO (130 kDa), three years later prompted purification of the 26-kDa TMEM147 as an additional, smaller ER-localized core component [1].

Beyond action in the Nicalin-NOMO-TMEM147 complex, which in addition to NODAL signaling also has a critical role in translocon function and thus ER-localized translation [5], TMEM147 interacts with the M3 muscarinic acetylcholine receptor (M3R), as has been shown with membrane-based yeast-2-hybrid detection systems, as well as with other class-I G-protein-coupled receptors, such as vasopressin-2 receptor and M1 muscarinic acetylcholine receptor (M1R). In this context, expression of TMEM147 in COS-7 cells inhibits trafficking of the receptor to the cell surface and additionally impairs receptor function, resulting in reduced levels of Ca^2+^ compared to control cells upon induction with the muscarinic agonist carbachol. Silencing of TMEM147 in H508 colon cancer cells, where TMEM147 and the M3R are normally co-expressed, causes an increase in the receptor molecules on the cell surface along with an overall increase in the properly folded receptors, implicating TMEM147 in both trafficking and folding of M3R [2].

Although *TMEM147* sequence conservation among organisms, its presence in a three-protein complex modulating the NODAL pathway and its interaction with M3R all point to the importance of TMEM147 and its interactions, details of TMEM147 function were until most recently largely unknown. To address this issue, our team has proven that TMEM147 not only localizes to ER membranes but also to the nuclear envelope (NE) of HeLa cells, as independently shown in immunofluorescence images labeled for either N-terminally FLAG-tagged and C-terminally GFP-tagged TMEM147 constructs [8], in line with incidental earlier observations in COS-7 cells [2]. Additionally, we have shown that depletion of TMEM147 causes transcriptional reduction of two ER-associated cholesterol biosynthesis pathway constituents. The first component is Lamin B receptor (LBR); a bifunctional protein best characterized for its N-terminal interaction with Lamin B and heterochromatin-related-proteins (HP1 and MeCR2) [9,10,11] and for its consequent role in tethering the transcriptionally inactive heterochromatin at the periphery of the NE [12]. However, in tandem to a chromatin-associated region extending throughout its C-terminus, LBR also contains a C-14 sterol reductase activity domain [13,14,15,16], showing extensive sequence similarity with evolutionary diversification for two other C-14 sterol reductases of the post-squalene biosynthetic pathway TM7SF2 and DHCR7 [12,17,18,19]. TMEM147 has been shown to physically interact with the C-terminal domain of LBR, which gave a first indication of a potential indirect regulatory role for TMEM147 in cholesterol synthesis. This notion was vindicated by our subsequent discovery of additional TMEM147 interaction with DHCR7 and of reduced transcript and protein levels for DHCR7 upon TMEM147 silencing [8].

Interactions of TMEM147 with NE components, such as with LBR, have implications beyond effects on cholesterol biosynthesis, including the localization of membrane components and NE structure. Specifically, in addition to transcriptional silencing of LBR, TMEM147 also effects a change in distribution for the remaining LBR protein, allowing detection of LBR in the proximal ER, in addition to the NE [8]. Additionally, TMEM147-silenced cells show significant changes in sphericity as a key parameter of nuclear shape, which is regulated by a complex network of pathways and factors [20]. These observations raise the question whether TMEM147, similar to its role in the NE, may also affect ER structure or abundance and position of ER integral membrane components. In this context, it is of interest that some components, such as TMEM147, may be shared between NE and ER, while others, such as LBR under normal conditions, are exclusively located in one of the compartments, despite the contiguity of ER and NE membranes and lumen.

A key example of shared ER and NE components is the reticulon (Rtn) family of highly conserved integral ER-membrane proteins with a characteristic wedge-shaped topological domain [21,22,23,24]. Rtn marks regions with high membrane curvature in both compartments, such as tubules in the smooth ER (sER) and the elongated highly bent membrane edges of rough ER (rER) sheets or matrices, as well as the developing nuclear pore complex of the NE [25]. The role of Rtn is well established in the ER, where its overexpression increases sER tubule length but reduces rER sheets [26], whereas Rtn depletion results in the expansion of peripheral sheets and the reduction in sER tubules and tubule branching [27,28]. Moreover, purified reticulons support membrane tubule generation from proteoliposomes in vitro [29]. Taken together, these data indicate that reticulons are both necessary and sufficient to induce tubule formation and to control the ratio of ER sheets to tubules [22,23].

A key example of components exclusive to the ER membrane is cytoskeleton linking membrane protein 63 (CLIMP63, formerly known as p63, official HGNC gene symbol *CKAP4*), which as a single-transmembrane ER protein binds microtubules with its N-terminus [30] and stabilizes ER luminal distance with its C terminus [22]. CLIMP63 overexpression results in sheet proliferation at the expense of tubules, whereas depletion of CLIMP63 unexpectedly does not affect the number of ER sheets, but instead decreases the rER luminal distance from ≈50 nm to ≈30 nm, comparable to that observed in lower organisms that lack CLIMP63 [31]. Furthermore, CLIMP63 overexpression diffuses sheets and the position of rER-ribosome-associated translocons throughout the cytoplasm [22], which is conversely not affected by CLIMP63 depletion [32] and which, moreover, establishes a tentative functional link to TMEM147 [5]. Highly expressed in the perinuclear rER, CLIMP63 is also detectable by advanced imaging in dynamic subdomains in peripheral ER tubules [24], where its expression and presence is associated with luminal elongation, while Rtn is associated with luminal branching, as Gao and co-workers revealed [28]. In either part of the ER, Rtn and CLIMP63 are therefore functional and topological opposites. Whereas Rtn is a marker of and essential for ER tubule, branch and edge formation as a major determinant of tubule:sheet ratio, CLIMP63 is a marker and facilitator of sheet formation and of unbranched ER areas [33]. Akin to LBR for the NE and regarding their role in the ER, CLIMP63 is both marker and structural determinant for the rER, while Rtn is both marker and structural determinant of the sER.

With our aim the comprehensive delineation of TMEM147 functions in the ER and cell-wide, we thus pursued two objectives. First, we set out to exploit our insights into CLIMP63 and Rtn function for an assessment of TMEM147 function in the ER. To this end, and in analogy to our previous analyses of TMEM147 and LBR concerning the NE, we investigated the effect of TMEM147 silencing on CLIMP63 and Rtn as markers and potential facilitators of TMEM147 ER action. Second, we set out to identify and summarize significantly TMEM147-associated protein networks and pathways as targets for future investigations and to address our currently limited understanding of TMEM147 multifunctionality, for which we employed comprehensive data mining and in silico analyses for reported TMEM147 upstream and downstream actors and interactors.

## 2. Results

### 2.1. Establishment of TMEM147 Silencing

For evaluation of TMEM147-dependent action we employed RNA interference (RNAi) to deplete *TMEM147* transcripts and protein, as previously reported [8]. Upon silencing of TMEM147, LBR is dramatically downregulated and its protein levels reduced by ≈90% [8], which allowed us to employ detection of LBR expression as a proxy for the confirmation of TMEM147 protein depletion (Figure 1a).

### 2.2. Silencing of TMEM147 Alters CLIMP63/RTN4 ER Labeling

To investigate the effect of TMEM147 silencing on ER area and composition, RNAi- and control-treated HeLa cells were stained with α-CLIMP63 and α-RTN4 antibodies to label the flattened and curved areas of the ER, respectively [22], LBR as a positive control for TMEM147 silencing in imaged cells and Hoechst 33342 as an LBR-independent nuclear size and DNA marker (Figure 1b,c). Upon TMEM147 silencing, ER labeling for both markers underwent pronounced elevation and expansion. CLIMP63 labeling expanded and increased, suggesting an increase in the flattened surface area and network density of ER (Figure 1b). In agreement with a denser ER network is also the observation of brighter RTN4 labeling (Figure 1c), suggesting an increase in the curved edges of the flat cisternae and tubular area, mainly at the periphery of the ER. The increase in both RTN4 and CLIMP63 labeling intensity and area additionally suggested an overall expansion of the ER density and surface area.

### 2.3. Statistical Analysis of ER Morphological Alterations in TMEM147-Silenced Cells

To quantify these observations concerning CLIMP63 and RTN4 and thus of ER composition and area after TMEM147-silencing, we conducted quantitative morphometric microscopy analyses for three independent silencing experiments and based on images acquired with identical, non-saturating-exposure settings (Figure 2a). Area analyses for a total of 457 cells (*n* = 232 silenced, *n* = 225 negative control) were performed for three independent fluorescence channels after labeling with α-CLIMP63 and α-RTN4 antibodies and Hoechst 33342 staining (Figure 2b).

*Mean Fluorescence Intensity (MFI)*. Analysis of MFI for DNA staining confirmed our previous finding that silencing of TMEM147 promotes a loss of chromatin compaction, with a significant decrease in the DNA signal after silencing (Figure 2c, *p* < 0.0001) [8]. By contrast, MFI for both ER markers, CLIMP63 and RTN4, significantly increased compared to that of untreated cells by 122.9% (*p* < 0.0001) and 60.3% (*p* < 0.0001), respectively, suggesting an increased density of the ER network and/or increased retention of both markers in ER membranes after TMEM147 silencing.

*Surface area analysis*. Corresponding analysis of nuclear surface area, ER surface area and ER flat and tubular domain surface areas revealed differential effects of TMEM147 silencing. While treatment had no effect on the surface area of the cell nucleus, it increased total ER surface area by 46.97% compared to control cells (*p* < 0.0001) (Figure 2d). Based on the universally fixed ratio of nuclear area to cell size [34,35], these MFI measurements indicated that TMEM147 silencing resulted in an almost 50% expansion of labeled ER area and a likely increase in ER density at unchanged cell size, with clear implications for the intracellular organization and transport of non-ER organelles. Moreover, the increase in ER area was not contributed in equal measure by both ER domains, with an area increase in flat ER cisternae of 100.13% (*p* < 0.0001) and with a more moderate increase in the tubular ER area of 41.39% (*p* < 0.0001) (Figure 2d).

*ER Domains Surface Area Ratios*. Differential area increases for flat and curved areas predictably resulted in a change in the area ratio for both ER domains compared to untreated cells (Figure 2e). While there was no difference in the ratio of curved area compared to the total ER area for treated vs. control-treated cells, the cisternal area increased by 34.22% (*p* < 0.0001) more than the total ER area and by 39.76% (*p* < 0.0001) more than the curved areas, clearly indicating a disproportionate expansion of flat areas upon silencing of TMEM147, in addition to an overall increase in ER area and density.

### 2.4. Data Mining to Compile TMEM147 Interactors

Taken together with our own previous observations for TMEM147 effects on LBR and cholesterol homeostasis, our present findings for its apparent inhibitory effect on ER size and density suggested a wide array of additional research targets for future investigation. In order to identify the most promising targets and experimental strategies for future investigations and to allow deeper insight into potential functional roles of TMEM147, we performed comprehensive datamining of interaction and association databases and of the literature, to identify all known TMEM147 interactions and associations. This revealed 98 interactors (Table 2; for full annotations see Appendix A), as input for subsequent network and pathway analysis.

### 2.5. Network Analyses of Reported TMEM147-Associated Proteins

We then used this input list of TMEM147 interactors, co-actors, targets and effectors for network analysis using String-db [36], in order to produce a comprehensive network of TMEM147-associated factors and pathways (Figure 3a). While identified from other repositories as interconnected, twelve of our list components were not associated by connections with any of the other 87 components. Likewise, comparison of our list-based analysis with identical analysis of TMEM147 as single input protein gave a network of only ten interactors (Figure 3b), excluding, e.g., LBR. Both observations indicated that while constantly updated and exceptionally powerful and versatile, String-db as an analysis tool cannot be a complete repository of interactions, e.g., for TMEM147, and benefits from the provision of manually curated input lists. 

Pathway-based analyses in String-db then revealed several pathway categories with statistically significant enrichment in the supplied list of TMEM147-associated proteins, including 25 GO Molecular Function terms, 21 GO Cellular Component terms, 7 UniProt Annotated Keyword terms, 6 GO Biological Process terms, 3 PubMed Reference Publications, 1 Reactome pathway, 1 Pfam Protein Domain (see Appendix A). As particularly informative for further analysis, the significantly enriched GO Molecular Function nodes were then exported to REVIGO [37] for content simplification.

### 2.6. Pathway Analysis for the TMEM147 Gene Network

Removal of redundant GO terms in REVIGO gave four functional groups of GO terms as a summary of a pathway-based analysis for all currently known TMEM147-associated proteins (Figure 4). The resulting non-redundant GO terms can be further summarized based on color coding provided by REVIGO, into our GO summary terms [A] *G-protein coupled receptor activity*; [B] *Transmembrane transport*; [C] *Oxidoreductase activity* and [D] *Ribosome binding* as given in Table 1, in back-annotation of corresponding GO terms, and thus GO summary terms, to each original protein identifier.

## 3. Discussion

### 3.1. Effect of TMEM147 Silencing on Rtn, CLIMP63 and ER Structure

In this study, we show that TMEM147 fundamentally affects the composition, the size and likely the structure and the density of the ER. While lowering DNA staining, silencing of TMEM147 significantly enhances labeling intensity and area of flat and curved ER areas, as measured by CLIMP63/CKAP4 and RTN4, respectively, as ER subdomain markers. TMEM147 silencing more substantially increases CLIMP63 labeling and thus additionally raises the ratio between flat and curved areas of the ER.

Importantly, both CLIMP63 and RTN4 are not inert markers of ER subdomains, but are dynamically regulated in their expression and distribution, with evidence of mutual exclusion in dynamic ER nanodomains [28]. Additional markers and, ideally, ultrastructural imaging will therefore be required to discern how far differences in brightness and extent of ER staining reported here reflect a difference in the ER membrane network or differential expression and/or retention in ER membranes of CLIMP63 and RTN4. Exclusion of both proteins from ER membranes as direct action by TMEM147 and conversely, their increased inclusion upon depletion of TMEM147 would be consistent with our observations here. In this context, marker-independent ultrastructural imaging would indicate the state of ER reorganization, while not revealing if any such change would be caused by or in turn causative of CLIMP63 and RTN4 enrichment in the ER for lowered TMEM147 levels. Whatever the causative mechanisms, increased ER area, and likely density, at identical cell size puts constraints on intracellular organization and transport of non-ER organelles, while facilitating ER luminal transport. This might be noticeable in transport defects, as could be evaluated in models, e.g., for secretory pathways or for in vitro neurogenesis, where transport represents a readily detectable bottleneck [38,39].

The differential effect of TMEM147 silencing on the expansion of tubular and cisternal compartments and, more immediately, on CLIMP63 and RTN4 abundance in the ER membrane, might be mirrored by corresponding TMEM147 effects in the NE. While based on our previous analyses we may exclude TMEM147 action on area, volume and ellipticity of the nucleus as key morphological parameters, TMEM147 silencing induced changes in nuclear sphericity and in LBR expression as nuclear effects [8]. It remains to be investigated whether in analogy to its action in the ER, TMEM147 depletion would also lead to an increase in structural NE membrane components. In the complex interplay of factors controlling nuclear size and shape [40], an increase in transient RTN4 in nascent nuclear pore complexes or, with exclusion of CLIMP63 from the NE [41], of functionally CLIMP63-analogous structural proteins, such as Nvj1 [42], might affect architecture and composition of the NE. Moreover, Rtn-dependent formation of ER tubules and formation of the NE are interlinked processes, in that NE assembly is dependent on ER tubules [43], but is encouraged by depletion of Rtn molecules [44]. In addition to interaction with LBR [8], putative interference with integral NE membrane components and with NE/ER membrane dynamics would therefore be an additional mechanism by which TMEM147 may effect changes in the nucleus.

### 3.2. Network Analysis for Additional TMEM147 Action

Despite a still limited number of studies, TMEM147 has already been implicated in an array of diverse nuclear and cytoplasmic actions and interactions, in line with other integral nuclear membrane proteins, such as the nesprin, emerin and SUN components of the LINC complex [45,46,47,48,49,50]. Manual curation of reported TMEM147 interactors, regulators and mediators thus revealed a wealth of protein factors in a multitude of pathways that show functional relationships with TMEM147. Statistical evaluation and simplification of the corresponding gene networks and pathways in turn highlighted specific highly connected proteins and prominent pathways that are informative for our growing understanding of TMEM147-related processes.

Pathway enrichment analysis, and in particular removal of redundant pathways post-analysis, crystallized key points of our current understanding of TMEM147 function. REVIGO-based simplification of findings confirmed *G-protein coupled receptor function* as one of four key clusters of pathways highlighted for TMEM147, tying in with TMEM147 regulation of muscarinic receptors [2]. Likewise, *Ribosome binding* fits the role of TMEM147 in the Nicalin-NOMO-TMEM147 complex and in the establishment of co-translational translocation in the ribosome associated translocon complex [5], whereas *Oxidoreductase activity* with the numerous protein components prompting its detection largely reflects the role of TMEM147 in sterol synthesis [8]. The identification of *Transmembrane transport*, with its associated factors mediating also ion transport across membranes, highlights an emerging aspect of TMEM147 biology, which has come into focus through the recent structural resolution of the Nicalin-NOMO-TMEM147 complex [5], specifically that beyond action in the rER translocon, the complex might dynamically interact with and facilitate transmembrane transport by hundreds of transmembrane channels. Besides its general implications for TMEM147 function, this insight is intriguing also with regard to the role of TMEM147 in the ER and suggests investigation of potential effects on Ca^2+^ uptake and release by the sER.

Creation of gene networks with our shortlist of factors in String-db expectedly established TMEM147 as the major hub protein of the overall network. More importantly, though possibly biased through the specific focus of a limited number of formally published or database-only studies, the resulting networks also gave indication of additional protein factors as possible key mediators of TMEM147 function. Accordingly, a high level of connectedness within the network indicated among others EBP, CD40, COX6B1, NFKB1, NHP2, STOML2 and YIPF6 as hub proteins. From these, the sterol reductase EBP fits into our understanding of TMEM147 as a regulator of sterol homeostasis [51]. Likewise, inclusion of the immune-related CD40 and NFKB1, with CD40 as a mediator of immune responses [52] and NFKB1 as a regulator of innate immunity [53], fits into our understanding of TMEM147 as an activator of proinflammatory NF-κB action [54]. However, other highly connected components are less expected, such as COX6B1and STOML2 with their roles in mitochondria. COX6B1 is a critical component of the cytochrome c oxidase complex and thus of mitochondrial function [55], whereas STOML2 is a facilitator of mitochondrial membranes and biogenesis [56]. Additional less expected highly connected protein factors are YIPF6, with a role in glycan synthesis and assembly of the Golgi apparatus [57], and NHP2, one of four protein factors required for telomere maintenance and ribosome biogenesis by association with nucleolar and telomerase RNAs [58].

### 3.3. Pointers for Future Investigation

Only eleven years after its discovery, TMEM147 has already been implicated in a wide array of physiological processes, many of which might be interconnected by common components or shared pathways. It is therefore difficult to discern primary from secondary or more distant downstream effects. For instance, analysis of protein networks and pathways has drawn new links to regulatory RNAs, the Golgi apparatus, glycans and mitochondria, which could be tested for their biological significance by knockout, knockdown or overexpression of TMEM147 and measurement of corresponding effects on ribosome, telomerase, Golgi, matrix and mitochondrial functions, respectively. However, TMEM147 is already linked to ribosome function through its action in the Nicalin-NOMO-TMEM147 complex [5], and effects on mitochondria and Golgi could also stem from our newly established TMEM147 action on the ER and from widely reported ER association with these organelles through membrane contact sites [38]. Once a TMEM147-associated phenotype has been established, discernment of corresponding primary and secondary effects will thus rely on further analyses, including the isolation of additional and possibly highly transient membrane-associated protein complexes. As for network analyses, the quandary of determining primary causality for a multifunctional protein also affects follow-up for our in vitro findings. If TMEM147 is inhibitory to density and expansion of smooth ER tubules, how much of its effect on sterol synthesis is mediated through interaction with LBR and how much through interference with the sER as a major intracellular compartment responsible for sterol synthesis? Indeed, TMEM147 effects on the ER alone touch on many biological phenomena and research fields with medical relevance. For instance, it is tantalizing to speculate in how far TMEM147 expression may be therapeutic in cancer by limiting RTN4 action [59]. Likewise, and given its profound effect on ER structure and size, TMEM147 expression might also limit the capacity for regular rER protein processing [60] and intracellular calcium signaling [61] (beyond its action on muscarinic acetylcholine receptors [2]), prevent useful or curb excessive ER expansion during ER stress situations [62], facilitate ER luminal transport or interfere with ongoing axonal dynamics in neurons [38,39,59]. Our analyses here indicate an expanding network of pathways and protein factors as a functional context for an array of future investigations.

## 4. Conclusions

Here we revealed through in silico analysis of protein networks that known TMEM147 activities affect four areas of molecular function and, based on established roles of highly interconnected proteins, propose additional, testable TMEM147 activities that extend beyond the function of already established direct interactors. Moreover, we showed experimentally that TMEM147 depletion increases ER area overall and flat areas in particular. Whether this occurs through a direct effect on ER domain markers and structural determinants CLIMP63 and RTN4 or through other interactions, the observation suggests wider action by TMEM147 on transmembrane proteins in both the ER and the NE, and turns TMEM147 into a putative regulator of many additional processes and into a potential therapeutic target for a range of diseases.

## 5. Materials and Methods

### 5.1. Cell Culture

HeLa (Kyoto; K) cells were used for all experimentation, cultured in complete medium (DMEM-GlutaMAX, 10% *v*/*v* heat-inactivated fetal bovine serum, penicillin/streptomycin at 100 U/mL and Sodium Pyruvate at 1 mM; all GIBCO/ThermoFisher, Waltham, MA, USA) and treated as previously reported [8].

### 5.2. RNA Interference

Cells were passaged in 60 mm culture dishes (Corning Inc., Corning, NY, USA), covered with sterile coverslips, to a cell density of 50–60%. 24 h later, cells were washed with DPBS (GIBCO/Thermo Fisher Scientific, Waltham, MA, USA) and the medium replaced with 4 mL of fresh complete medium. The siRNA oligonucleotide mixture were prepared by adding 8 μL of 40 nM siRNA oligonucleotides (#109789 Silencer^®^ Pre-designed siRNA, 5′-GGCGGCAUCUAUGACUUCATT-3′, Thermo Fisher Scientific/Ambion Inc., Austin, TX, USA), or 8 μL water for negative controls, and 12 μL INTERFERin transfection agent (Polyplus Transfection, Vectura, Illkirch-Graffenstaden, France) to 400 μL OptiMEM (GIBCO/Thermo Fisher Scientific, Waltham, MA, USA) before vortexing and 10 min incubation at room temperature, according to the transfection reagent manufacturer’s recommendations. At t = 0 h, the mixture was added dropwise to the cells with gentle agitation. At t = 24 h, the medium was changed as required and at t = 72 h coverslips were harvested for immunofluorescence (IF) analysis.

### 5.3. Immunofluorescence

All incubation and handling steps were at room temperature, unless indicated otherwise.

#### 5.3.1. Antibodies Used

Antibodies and dilutions for immunofluorescence were, as primary antibodies, mouse αCLIMP63 (dilution 1:1000G1/296, ABS669-0100 Enzo Life Sciences, Farmingdale, NY, USA), rabbit αRTN4 (dilution 1:300, ab47085, Abcam, Cambridge, UK), rabbit αLBR (dilution 1:300, ab32535, Abcam, Cambridge, UK), and as secondary antibodies goat αRabbit AlexaFluor488 (1:1000, IgG/A-11034, Molecular Probes/Invitrogen/Thermo Fisher Scientific, Waltham, MA, USA) and donkey αMouse AlexaFluor555 (1:1000, IgG/A-31570, Molecular Probes/Invitrogen/Thermo Fisher Scientific, Waltham, MA, USA). Hoechst 33342 was used as a DNA counterstain at 0.5 μg/mL (B2261, Sigma-Aldrich, St. Louis, MO, USA).

#### 5.3.2. Cell Fixation

Fixation solution was prepared as paraformaldehyde (PFA) 4% (*w*/*v*) in PHEM pH 8.0 (30 mM HEPES, 60 mM Pipes pH 6.9, 10 mM EGTA, 2 mM MgCl_2_) and permeabilization solution as 0.5% (*v*/*v*) triton-X in PHEM (all Sigma-Aldrich, St. Louis, MO, USA). Coverslips were placed in a 12-well plate containing fixation solution, followed by 10 min of incubation, replacement of the fixation solution with permeabilization solution for 5 min of incubation, and three 3 min washes of coverslips with PHEM. After application of 50 mM NH_4_Cl in DPBS as quenching solution and 10 min incubation, coverslips were stored in DPBS until immunofluorescence staining.

#### 5.3.3. Immunofluorescence Staining and Mounting

Coverslips bearing fixed cells were placed on PARAFILM^®^ (Sigma-Aldrich, St. Louis, MO, USA) and incubated with blocking solution (2% (*w*/*v*) FBS, 2% (*w*/*v*) BSA, 0.2% (*w*/*v*) fish skin gelatin, in DPBS) for 1 h, before its replacement by the appropriate dilution of primary antibody or a mixture of primary antibodies in 5% blocking solution in DPBS and incubation for 16–18 h at 4 °C. After three 3 min washes with washing solution (0.1% (*v*/*v*) Tween^®^ 20 in PHEM), the appropriate secondary antibody or antibody mixture in 5% blocking solution in DPBS was applied and left to incubate at room temperature for 1 h, followed by two 3 min washes in washing solution and one 3 min wash in DPBS, before mounting of the dry coverslip using Dako Fluorescence Mounting Medium (Agilent/Dako, Santa Clara, CA, USA)

### 5.4. Microscopy

Microscopy was performed on a Carl Zeiss Axiovert 200 Μ with a Plan-Apochromat 63×/1.4NA oil lens and wavelength filter sets for detection of Hoechst 33342 (450 nm), Alexa Fluor 488 (520 nm) and Alex Fluor 555 (568 nm) The images were acquired using the high sensitivity digital camera AxioCam HRc for multicolor fluorescent images along with the Zeiss software Axiovision, v4.8.2.SP2. All same-figure images were taken at identical exposure and filter settings for comparability of intensities. Images were processed with uniform and unbiased application of contrast and brightness settings in Photoshop and Illustrator (Adobe Inc., San Jose, CA, USA).

### 5.5. Morphometric Analysis

Using the standard microscopy setup, ER alterations after TMEM147 silencing were measured after staining for CLIMP63 and RTN4 for a total of 457 HeLa cell images, including 225 control-silenced and 232 TMEM147-silenced cells, from three independent experiments and with identical acquisition settings across control and corresponding silenced samples. The pixel area of staining for CLIMP63 (corresponding to flattened ER cisternae) and RTN4 (corresponding to ER tubules) was measured in ImageJ v1.52p by manually selecting the boundaries of each cell as a region of interest (ROI). For each cell and fluorescence channel, thresholding algorithms were applied to determine the corresponding area, and the total ER area was calculated as the union of both channels without double-counting areas of overlap.

### 5.6. Statistical and Bioinformatics Evaluation

#### 5.6.1. Network Analyses

Reported TMEM147 targets, direct, indirect and same-complex interactors, co-regulated and co-expressed proteins were identified and manually curated for redundancy and spurious or erroneous database entries based on the following repositories: APID [63], BioGRID, GeneMania [64], GPS-Prot [65], Human Protein Reference Database [66,67], IntAct EMBL-EBI [68], HuRi [69], PubMed, Pathway Commons [70], PINA [71,72], Reactome [73], STRING [36], with identifiers and annotations from UniProt [74].

The resulting list of 98 protein factors plus TMEM147 was used as a list of input proteins in String-db.org 11.0b (STRING, Available online: https://version-11-0b.string-db.org (accessed on 28 July 2021)) [36], applying as settings full network analysis, network edges representing evidence, using all active interaction sources, applying 0.150 as the required interaction score and displaying only the query proteins and no additional layers of secondary interactors (for Figure 3b, including the first shell of interactors for illustration). The network output images were manually curated to align unlinked protein factors outside the network, to apply the official HUGO gene symbol where String-db applied alternative protein names and to include the legend. 

#### 5.6.2. Pathway Reduction

Significant Gene Ontology Molecular Function nodes derived from String-db were imported with their corresponding false discovery rate (FDR) values to the Reduce and Visualize Gene Ontology (REVIGO, Available online: http://revigo.irb.hr (accessed on 14 August 2021)) tool [37], in order to remove redundant GO terms and interpret more readily the underlying enrichment analysis. The settings applied were the choice of Small (0.5) for the resulting list, the provision of P value (as the most suitable choice) for associated values, the removal of obsolete GO terms, the choice of *Homo sapiens* as target species and the choice of the default SimRel semantic similarity measure. Of the resulting output, TreeMap and the table of non-redundant GO terms were included in Figure 3, whereas corresponding Scatterplot and Interactive Function graphs are available as Appendix A.

#### 5.6.3. Summary Statistics and Groupwise Comparisons

Summary statistics and groupwise comparisons for image analyses based on DNA, CLIMP63 and RTN4 fluorescence measurements after TMEM147 silencing were based on composite images for a total of 457 HeLa cells. Box-and-whisker diagrams displaying all values were based on raw data for all cells, whereas bar charts and groupwise comparisons were based on data after outlier removal using the 1.5x IQR rule [75], resulting in evaluable data for 362 HeLa cells (n = 177 control, n = 185 silenced). These data were tested for statistical significance between untreated and treated cells by MANOVA analysis using Pillai’s test at a statistical significance α = 5% in Excel (Microsoft Office, 2016, Redmond, WA, USA) and XLSTAT (Addinsoft, Paris, France).

## Figures and Tables

**Figure 1 ijms-22-10231-f001:**
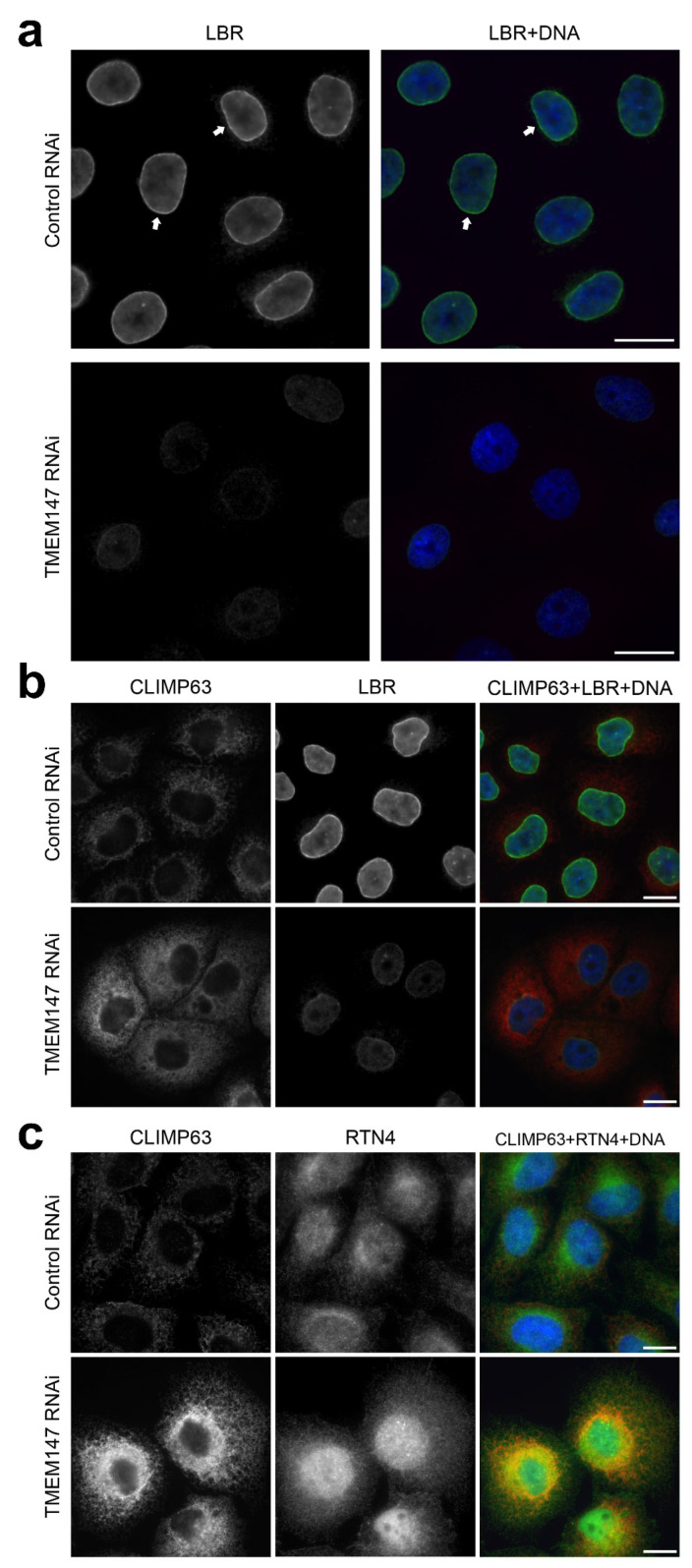
Immunofluorescence imaging of TMEM147 silencing in HeLa cells. (**a**) Cells stained for LBR and DNA show downregulation of LBR expression in response to TMEM147 silencing, in line with [8]; arrows indicate the nuclear rim on the inner nuclear membrane; (**b**) cells stained for CLIMP63, LBR and DNA; (**c**) Cells stained for CLIMP63, RTN4 and DNA. For (**b**,**c**), TMEM147-silenced cells show elevated fluorescence intensity and area for the respective ER marker. Size markers indicate 20 μm.

**Figure 2 ijms-22-10231-f002:**
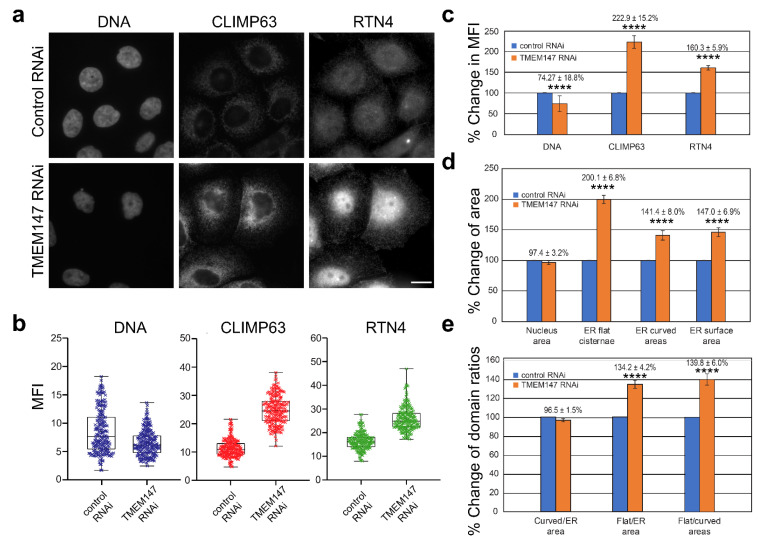
Statistical evaluation of immunofluorescence imaging after RNAi treatment. (**a**) Representative images employed for fluorescence and area quantifications; (**b**) box-and-whisker diagrams of mean fluorescence intensities for DNA, CLIMP63 and RTN4, as indicated; (**c**) bar chart of mean fluorescence intensity (MFI) percentage changes in DNA, CLIMP63 and RTN4 signals for TMEM147 vs. control silencing; (**d**) Bar chart of organelle surface area percentage changes of nucleus (measured as DNA), ER flat cisternae (measured as CLIMP63), ER curved areas (measured as RTN4) and total ER surface area (measured as the overlap-corrected union of CLIMP63 and RTN4 areas) for TMEM147 vs. control silencing; (**e**) bar chart of percentage changes in ER domain ratios, corresponding to (**d**). Error bars indicate the standard deviation of the population. Size markers indicate 20 μm. **** indicates *p* < 0.0001.

**Figure 3 ijms-22-10231-f003:**
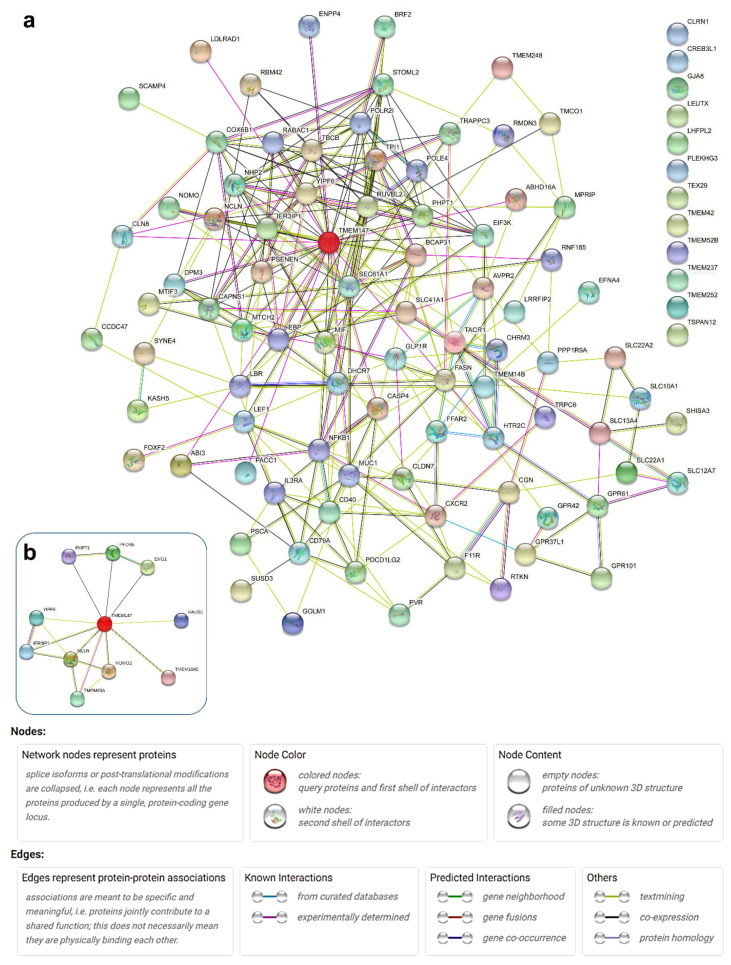
Network analysis for TMEM147-associated proteins. (**a**) Primary network of proteins based on our manually curated list of known interactions and reports for TMEM147 (indicated by transparent red overlay) as analyzed in String-db 11.0b. Proteins without interaction in the String-db 11.0b database are aligned at the top right. (**b**) Primary network of TMEM147-associated proteins as held in String-d 11.0b, based on TMEM147 as input. The color code for underlying database information used by String-db is given in the in-figure legend. Of note, no white nodes are present, because network analysis was restricted to query proteins (**a**) or the first shell of interactors (**b**).

**Figure 4 ijms-22-10231-f004:**
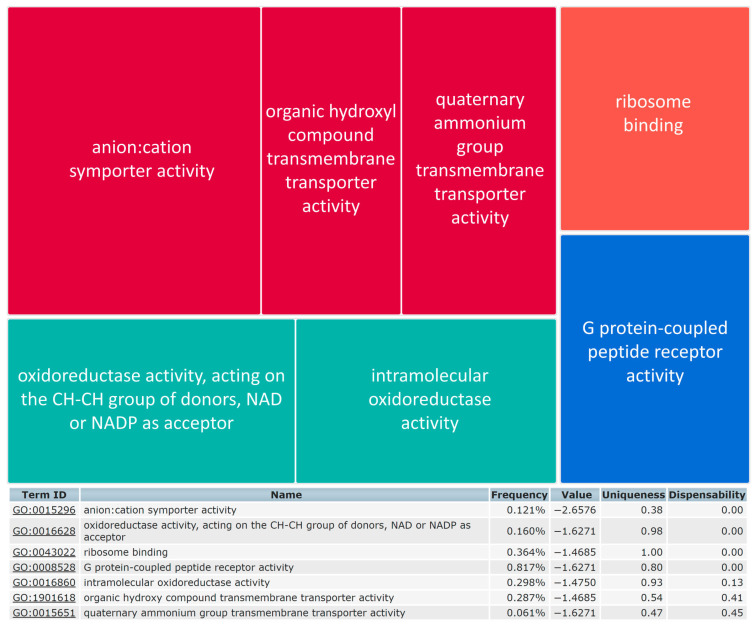
Reduction in pathway complexity for TMEM147. Reduction in Gene Ontology Molecular Function terms for TMEM147 interactors based on enriched Gene Ontology Molecular Function terms identified by String-db 11.0b, with target being a “Small” output list in REVIGO. Alternatively, non-redundant GO terms are provided by REVIGO in Scatterplots and Interactive Maps, as shown in Appendix A. For clarity, fonts for TreeMap and table outputs have been enhanced compared to the original REVIGO display.

**Table 1 ijms-22-10231-t001:** TMEM147 variants in *H. sapiens*. Range information for Variants 2 and 3 indicates corresponding amino acids for Variant 1; nt—nucleotides; aa—amino acids.

Variant	mRNA Length	Protein Length	Accession No.	Note
1	868 nt	224 aa	NP_116024.1NM_032635.4	Main isoform
2	939 nt	175 aa (aa 50–224)	NP_001229526.4NM_001242597.2	Compared to Variant 1 with truncated N terminus and identical C terminus, by inclusion of an additional exon in its extended 5′ UTR and initiating translation at a downstream, in-frame start codon
3	646 nt	150 aa (aa 1–70, aa 144–224)	NP_001229527.1NM_001242598.2	Compared to Variant 1 with identical N and C terminus, but lacking two consecutive exons in the coding region; shortest variant

**Table 2 ijms-22-10231-t002:** Reported TMEM147 associations and interaction. This manually curated, non-redundant list of proteins as retrieved from twelve different repositories is the basis for network and pathway enrichment analyses. The input list of identifiers for String-db and an annotated version of this table are provided as Appendix A.

Gene Symbol ^a^ [String-db ID] ^b^	Uniprot ID ^c^	Interaction Type ^d^	GO IDs ^e^	GO Group/ Context ^f^	Protein ^g^	Source Db ^h^
*ABHD16A*	O95870	A			Phosphatidylserine lipase ABHD16A	A,B,C,D,E,G,F,J
*ABI3*	Q9P2A4	A			ABI gene family member 3	A,B,D,E,G,F,I,J
*AVPR2*	P30518	B	4888, 8528, 38023, 4930	A	Vasopressin V2 receptor	C
*BCAP31*	P51572	F			B-cell receptor-associated protein 31	C
*BRF2*	Q9HAW0	B			Transcription factor IIIB 50 kDa subunit	A,D,F,I,J
*CAPNS1*	P04632	F			Calpain small subunit 1	C
*CASP4*	P49662	B			Caspase-4	A,D,F,J
*CCDC47*	Q96A33	A			PAT complex subunit CCDC47	H
*CD40*	P25942	A	4888, 38023	A	Tumor necrosis factor receptor superfamily member 5	A,B,G,F,I,J
*CD79A*	P11912	A	4888, 38023	A	B-cell antigen receptor complex-associated protein alpha chain	A,B,G,F,I,J
*CGN*	Q9P2M7	B			Cingulin	A
*CHRM3*	P20309	A	4888, 38023, 8227, 4930	A	Muscarinic acetylcholine receptor M3	C,H
*CLDN7*	O95471	A			Claudin-7	A,B,G,F,I,J
*CLN8*	Q9UBY8	A			Protein CLN8	A,B,D,H,J
*CLRN1*	P58418	A			Clarin-1	A,B,D,H,J
*COX6B1*	P14854	F	22890, 15077	B	Cytochrome c oxidase subunit 6B1	C
*CREB3L1*	Q96BA8	A			Cyclic AMP-responsive element-binding protein 3-like protein 1	A,B,D,G,F,I,J,K
*CXCR2*	P25025	A	4888, 8528, 38023, 4930	A	C-X-C chemokine receptor type 2	A,B,G,F,I,J,K
*DHCR7*	Q9UBM7	A	16628	C	7-dehydrocholesterol reductase	H
*DPM3*	Q9P2X0	A			Dolichol-phosphate mannosyltransferase subunit 3	A,B,G,F,I,J
*EBP*	Q15125	A	4888, 38023, 16860, 16863	A,C	3-beta-hydroxysteroid-Delta(8),Delta(7)-isomerase	A,B,G,F,I,J,K
*EFNA4*	P52798	A	4888, 38023	A	Ephrin-A4	A,B,G,F,I,J
*EIF3K*	Q9UBQ5	F	43022	D	Eukaryotic translation initiation factor 3 subunit K	C
*ENPP4*	Q9Y6X5	A			Bis(5′-adenosyl)-triphosphatase ENPP4	A,B,G,F,J
*F11R*	Q9Y624	A			Junctional adhesion molecule A	A,B,G,F,J,K
*FASN*	P49327	G	16628	C	Fatty acid synthase	B
*FFAR2*	O15552	A	4888, 38023, 4930	A	Free fatty acid receptor 2	A,B,G,F,J,K
*FOXF2*	Q12947	E			Forkhead box protein F2	I
*GJA8*	P48165	A			Gap junction alpha-8 protein	A,B,G,F,J
*GLP1R*	P43220	A	4888, 8528, 38023, 4930	A	Glucagon-like peptide 1 receptor	A,B,D,G,H,J
*GOLM1*	Q8NBJ4	A			Golgi membrane protein 1	A,B,G,F,J
*GPR101*	Q96P66	A	4888, 38023, 8227, 4930	A	Probable G-protein coupled receptor 101	A,B,G,F,J
*GPR37L1*	O60883	A	4888, 8528, 38023, 4930	A	G-protein coupled receptor 37-like 1	A,B,G,F,J,K
*GPR42*	O15529	A	4888, 38023, 4930	A	G-protein coupled receptor 42	B,G,F,I,J
*GPR61*	Q9BZJ8	A	4888, 38023, 4930	A	G-protein coupled receptor 61	A,B,G,F,J
*HTR2C*	P28335	A	4888, 38023, 8227, 4930	A	5-hydroxytryptamine receptor 2C	A,B,D,G,F,J
*IER3IP1*	Q9Y5U9	C			Immediate early response 3-interacting protein 1	L
*IL3RA*	P26951	A	4888, 38023	A	Interleukin-3 receptor subunit alpha	A,B,G,F,J,K
*KASH5* [*CCDC155*]	Q8N6L0	A			Protein KASH5	A,B,G,F,I,J
*LBR*	Q14739	A	16628	C	Delta (14)—Sterol Reductase LBR	H
*LDLRAD1*	Q5T700	A			Low-density lipoprotein receptor class A domain-containing protein 1	A,B,D,G,F,J
*LEF1*	Q9UJU2	E	38023	A	Lymphoid enhancer-binding factor 1	I
*LEUTX*	A8MZ59	A			Paired-like homeodomain transcription factor LEUTX	A,B,G,F,I,J
*LHFPL2*	Q6ZUX7	A			LHFPL tetraspan subfamily member 2 protein	A,B,G,F,J
*LRRFIP2*	Q9Y608	B			Leucine-rich repeat flightless-interacting protein 2	A
*MIF*	P14174	F	16860, 16863	C	Macrophage migration inhibitory factor	C
*MPRIP*	Q6WCQ1	B			Myosin phosphatase Rho-interacting protein	A
*MTCH2*	Q9Y6C9	C			Mitochondrial carrier homolog 2	L
*MTIF3*	Q9H2K0	A	43022	D	Translation initiation factor IF-3, mitochondrial	A,B,G,F,I,J,K
*MUC1*	P15941	A			Mucin-1	A,B,G,F,I,J
*NCLN*	Q969V3	A			Nicalin	C,H,L
*NFKB1*	P19838	A			Nuclear factor NF-kappa-B p105 subunit	H
*NHP2*	Q9NX24	F			H/ACA ribonucleoprotein complex subunit 2	C
*NOMO*		A			NODAL modulator	H
*PACC1* [*TMEM206*]	Q9H813	A			Proton-activated chloride channel	A,B,D,G,F,J
*PDCD1LG2*	Q9BQ51	A			Programmed cell death 1 ligand 2	A,B,G,F,I,J
*PHPT1*	Q9NRX4	F			14 kDa phosphohistidine phosphatase	L
*PLEKHG3*	A1L390	B			Pleckstrin homology domain-containing family G member 3	A
*POLE4*	Q9NR33	F			DNA polymerase epsilon subunit 4	C
*POLR2I*	P36954	F			DNA-directed RNA polymerase II subunit RPB9	C
*PPP1R9A*	Q9ULJ8	B			Neurabin-1	A
*PSCA*	O43653	A			Prostate stem cell antigen	A,B,G
*PSENEN*	Q9NZ42	F			Gamma-secretase subunit PEN-2	L
*PVR*	P15151	A	38023	A	Poliovirus receptor	A,B,G,F,J,K
*RABAC1*	Q9UI14	F			Prenylated Rab acceptor protein 1	C
*RBM42*	Q9BTD8	F			RNA-binding protein 42	C
*RMDN3*	Q96TC7	A			Regulator of microtubule dynamics protein 3	A,B,G,F,J
*RNF185*	Q96GF1	A			E3 ubiquitin-protein ligase RNF185	A,B,G,F,J
*RTKN*	Q9BST9	B			Rhotekin	A
*RUVBL2*	Q9Y230	F			RuvB-like 2	C
*SCAMP4*	Q969E2	A			Secretory carrier-associated membrane protein 4	A,B,G,J
*SEC61A1*	P61619	A	43022	D	Protein transport protein Sec61 subunit alpha isoform 1	H
*SHISA3*	A0PJX4	A			Protein shisa-3 homolog	A,B,G,F,J
*SLC10A1*	Q14973	A	15294, 15370, 15291, 15081, 46873, 1901618, 22890, 15077	B	Sodium/bile acid cotransporter	A,B,G,F,J,K
*SLC12A7*	Q9Y666	A	15296, 15377, 15294, 15291, 46873, 22890, 15077	B	Solute carrier family 12 member 7	A,B,G,F,J
*SLC13A4*	Q9UKG4	A	15296, 15373, 15294, 15370, 15291, 15081, 46873, 22890, 15077	B	Solute carrier family 13 member 4	A,B,G,F,I,J,K
*SLC22A1*	O15245	A	15296, 15373, 15377, 5277, 5330, 5334, 8513, 15294, 15370, 15651, 15291, 15081, 46873, 1901618, 22890, 15077	B	Solute carrier family 22 member 1	A,B
*SLC22A2*	O15244	A	15296, 15373, 15377, 5277, 5330, 5334, 8513, 15294, 15370, 15651, 15291, 15081, 46873, 1901618, 22890, 15077	B	Solute carrier family 22 member 2	A,B
*SLC41A1*	Q8IVJ1	A	15291, 15081, 46873, 22890, 15077	B	Solute carrier family 41 member 1	H
*STOML2*	Q9UJZ1	F			Stomatin-like protein 2, mitochondrial	C
*SUSD3*	Q96L08	A			Sushi domain-containing protein 3	A,B,G,F,I,J
*SYNE4*	Q8N205	A			Nesprin-4	A,B,D,G,F,I,J
*TACR1*	P25103	A	4888, 8528, 38023, 4930	A	Substance-P receptor	A,B,G,F,I,J,K
*TBCB*	Q99426	F			Tubulin-folding cofactor B	C
*TEX29*	Q8N6K0	A			Testis-expressed protein 29	A,B,G,F,J
*TMCO1*	Q9UM00	A	46873, 22890	B,D	Calcium load-activated calcium channel	H
*TMEM14B*	Q9NUH8	A			Transmembrane protein 14B	A,B,G,F,J
*TMEM237*	Q96Q45	A			Transmembrane protein 237	A,B,G,F,J
*TMEM248*	Q9NWD8	A			Transmembrane protein 248	A,B,G,F,I,J
*TMEM252*	Q8N6L7	A			Transmembrane protein 252	A,B,G,F,I,J
*TMEM42*	Q69YG0	A			Transmembrane protein 42	A,B,G,F,J
*TMEM52B*	Q4KMG9	A			Transmembrane protein 52B	A,B,G,F,I,J
*TPI1*	P60174	F	16860	C	Triosephosphate isomerase	C
*TRAPPC3*	O43617	F			Trafficking protein particle complex subunit 3	C
*TRPC6*	Q9Y210	H	46873, 22890	B	Short transient receptor potential channel 6	B,H,J
*TSPAN12*	O95859	A			Tetraspanin-12	A,B,G,F,J
*YIPF6*	Q96EC8	C			Protein YIPF6	L

^a^ Official gene symbol according to the HUGO Gene Nomenclature Committee (HGNC); ^b^ the associated STRING ID used in STRING analysis, if different from the official gene symbol; ^c^ protein UniProt ID; ^d^ nature of discovered TMEM147 interactions: [A] direct physical interaction: the interaction is at the protein–protein level; [B] physical interaction: co-existence in a stable complex; [C] indirect physical interaction: both proteins have a common direct physical interactor; [D] functional association: the combined function with TMEM147 affects the corresponding phenotype; [E] expression regulation: the interactor controls the expression of TMEM147; [F] genetic: significant co-regulation, protein expression dependency or intercellular co-localization with TMEM147; [G] positive modifier: the phenotype is more severe when both are defective; [H] negative modifier: the phenotype is milder when both are defective. ^e^ Gene Ontology ID with format GO:###### after removal of “GO:” and leading zeros for the sake of compactness; ^f^ representative non-redundant Gene Ontology category according to REVIGO: [A] G-protein coupled receptor activity; [B] Transmembrane transport; [C] Oxidoreductase activity; [D] Ribosome binding; ^g^ protein name as found in UniProt; ^h^ query sources: [A] APID; [B] BioGRID; [C] GeneMania; [D] GPS-Prot; [E] Human Protein Reference Database; [F] IntAct EMBL-EBI; [G] HuRi; [H] PubMed; [I] Pathwaycommons; [J] PINA3; [K] Reactome; [L] STRING.

## Data Availability

Not applicable.

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
