# Peer review of "Regulation of ER Composition and Extent, and Putative Action in Protein Networks by ER/NE Protein TMEM147"

_ijms, 2021, doi:10.3390/ijms221910231_

Round 1

Reviewer 1 Report

The authors investigated the protein network and pathway analyses based on a comprehensive compilation of TMEM147 interactors, targets, and co-factors because of its exceptional emerging multifunctionality. Significant pathways that have been algorithmically simplified reflect the role of TMEM147 in ribosome binding, oxidoreductase activity, G protein-coupled receptor activity, and transmembrane transport, while protein factors and networks analysis identifies hub proteins and corresponding pathways as potential targets of TMEM147 action and future functional studies. Overall, it is an important study and should be considered for publication once the issues have been resolved.

  • The abstract is not clear. Please add the aim and objective of the MS.
  • Please add the objective of MS at the end of the introduction part.
  • Please speculate about the reasons for the obtained results. The discussion needs to improve.
  • Please add the Conclusion. There is no conclusion in the MS. In Conclusion, the authors should add the potential practical application.
  • The paper contains a typo and grammatical error which should be fixed.

Reviewer 2 Report

This research work is well-planned, well-designed, using bioinformatics tools along with experimental results the authors provided enough pieces of evidence to prove their hypothesis. However, I have the following minor suggestions,

  1. Line 293, please add a space before the [57].
  2. Reference 3 and 4 end with a comma.
  3. Supplementary Figure 4 texts are too small to see.

Round 2

Reviewer 1 Report

Requested corrections were completed.